# Bridging the Gap between Simulation and Real Autonomous UAV Flights in Industrial Applications

Rafael Perez-Segui *, Pedro Arias-Perez , Javier Melero-Deza , Miguel Fernandez-Cortizas , David Perez-Saura and Pascual Campoy

Computer Vision and Aerial Robotics Group, Centre for Automation and Robotics (C.A.R.),
Universidad Politécnica de Madrid (UPM-CSIC), 28006 Madrid, Spain; david.perez.saura@upm.es (D.P.-S.)
* Correspondence: r.psegui@upm.es

**Abstract:** The utilization of autonomous unmanned aerial vehicles (UAVs) has increased rapidly due to their ability to perform a variety of tasks, including industrial inspection. Conducting testing with actual flights within industrial facilities proves to be both expensive and hazardous, posing risks to the system, the facilities, and their personnel. This paper presents an innovative and reliable methodology for developing such applications, ensuring safety and efficiency throughout the process. It involves a staged transition from simulation to reality, wherein various components are validated at each stage. This iterative approach facilitates error identification and resolution, enabling subsequent real flights to be conducted with enhanced safety after validating the remainder of the system. Furthermore, this article showcases two use cases: wind turbine inspection and photovoltaic plant inspection. By implementing the suggested methodology, these applications were successfully developed in an efficient and secure manner.

**Keywords:** aerial robotics; Aerostack2; industrial inspections; photovoltaic plant; ROS 2; Sim2Real; UAV; wind turbine

## 1. Introduction

Aerial robotics has become a revolutionary technology in various industries, providing multitude of benefits by its integration into the industrial inspection processes. Unmanned Aerial Vehicles (UAVs) excel at accessing hard-to-reach areas quickly and accurately while ensuring the safety of industrial equipment [1]. These advantages are primarily attributed to its ability to navigate through terrain and tight spaces with precision and speed.

For these applications, the use of simulation has proven to be determining in the development of autonomous algorithm. Its utilization offers several crucial advantages, starting with the ability to test thoroughly and validate algorithms prior to their deployment in safe ways [2]. By operating within a virtual environment, engineers can easily detect and resolve any issues without incurring high costs or endangering the drone's integrity, the industrial facilities or workers.

Furthermore, simulation enables engineers to assess algorithms performance across diverse scenarios, encompassing varying weather patterns, lighting conditions, and obstacles. These capabilities allow for a comprehensive evaluation of the algorithm's robustness and adaptability, providing valuable insights for further improvement.

Another notable benefit of simulation in autonomous algorithms testing is the ability to anticipate and address potential safety issues before deployment. Autonomous drones involve potential safety hazards, including collisions with objects and people. Simulation allows researchers and engineers to anticipate and address potential safety issues, ensuring that the algorithm operates within safe parameters.

Once a design has been optimized in simulation, it can be tested in real flight experiments to validate its performance and safety. But carrying out experiments in certain

environments can be risky. For example, industrial installations contain expensive and sensitive equipment, and accidents can result in costly damage. In such cases, using simulation data to inform and guide real-world experiments allows testing the behavior of the robotic system far from the final installations, reducing the risk of costly accidents and increasing safety during development.

Therefore, in the state of the art, there arises the need for a generic methodology to develop all kinds of industrial applications in a safe, efficient, and cost-effective way. This will enable the automation of industrial processes, such as the inspection of wind turbines or photovoltaic plants through aerial robotics.

This work presents two main contributions:

- It proposes a generic methodology for the development of industrial applications using aerial robots. It demonstrates the methodology used in two different and relevant use cases, where simulation data are used to guide and validate real flight experiments. This approach allows for a safer and more efficient testing process, as potential issues and risks can be identified and mitigated in the simulation environment before moving on to final environment experiments.
- It explains how the methodology uses Aerostack2 framework [3] for the development process of a real industrial application, from simulation to real implementation, which provides a comprehensive and flexible solution for the design and implementation of autonomous UAVs.

The remainder of this paper is organized as follows. In Section 2, it presents most relevant related work to this paper, highlighting main simulators and aerial stacks, besides recent autonomous aerial industrial inspections. The proposed methodology will be described in Section 3. Section 4 explains how the transition from simulation to the real world has been carried out. Following, Sections 5 and 6 show two different industrial scenarios, wind turbine and photovoltaic plant inspection, where this methodology has been applied. Section 7 presents the results obtained in the experiments carried out for each industrial applications. Finally, Section 8 concludes this work while commenting on possible future work. The Acknowledgements and References follow Section 8.

## 2. Related Work

The utilization of drones in various industrial applications has been widely observed, as evidenced by numerous studies [4]. However, while drones are commonly used in these applications, only a subset of them is capable of conducting autonomous inspections. For instance, in the case of [1,5], the development process involves an initial phase of simulation followed by a transition to real environments. Similarly, Ref. [6] and similar studies opt for developing in controlled environments that closely resemble the final operational settings albeit at a higher cost for creating such environments. In conclusion, the existing state of the art underscores the pressing need for an efficient and cost-effective methodology to test algorithms in settings that are distant from industrial facilities, thereby mitigating potential risks associated with real-world testing.

Regarding the simulators used, a big number of them are available and can be used in the initial phase for autonomous UAV testing, each of them presenting the following specific features that are discussed for current purposes.

RotorS [7] is a modular Micro-Aerial Vehicle (MAV) simulation framework built on Gazebo [8], which allows a quick start to perform research on MAVs. The simulator was designed in a modular way so that different controllers and state estimators can be used interchangeably, while incorporating new MAVs is reduced to a few steps. The provided controllers can be adapted to a custom vehicle by simply changing a parameter file. Different controllers and state estimators can be compared with the provided evaluation framework. All components were designed to be analogous to their real-world counterparts. This allows the usage of the same controllers and state estimators, including their parameters, in the simulation as on the real MAV.

AirSim [9] is a photo-realistic simulator built on Unreal Engine that offers physically and visually realistic simulations. It includes a physics engine that can operate at a high frequency for real-time hardware-in-the-loop (HIL) simulations with support for popular protocols (e.g., MavLink). The simulator is designed from the ground up to be extensible to accommodate new types of vehicles, hardware platforms and software protocols.

FlightGoggles [10] is an open-source photo-realistic sensor simulator for perception-driven robotic vehicles. It consists of two separate components, the photo-realistic rendering engine built on Unity and a quadrotor dynamics simulation engine. It also provides an interface with real-world aircrafts for image and data processing.

Flightmare [11]: Like FlightGoggles, Flightmare is a flexible modular quadrotor simulator composed of two main components: a configurable rendering engine built on Unity and a flexible physics engine for dynamics simulation. Those two components are totally decoupled and can run independently from each other. In addition, it also provides an interface with the Gazebo simulator.

In conclusion, after examining several simulators for autonomous drone flights, it can be concluded that they do not provide a direct pathway from simulation to real-world drone operations. Rather, they focus on specific components or aspects of drone flights, such as a simulation training environment, testing of a specific platform or image processing algorithms. These simulators are valuable tools for training and testing autonomous drone systems in a controlled environment, but they do not necessarily prepare them for the complex and unpredictable realities of real-world drone operations. In addition, they do not facilitate the simulation for any platform, specializing in several with specific characteristics.

In the domain of software options for developing industrial inspection applications using unmanned aerial systems (UASs), a comparative analysis is presented in Table 1, which was adapted from [3]. The selection process focused on identifying a framework capable of facilitating simulation and implementation beyond laboratory settings. Moreover, due to the specific requirements of inspection tasks, the chosen framework needed to support multi-agent functionality and be compatible with diverse robotic platforms. Based on these considerations, the study concludes that Aerostack2 satisfies the aforementioned criteria, being the only one that meets the requirements of being modular, multi-agent, and multi-platform, facilitating the development of the proposed methodology.

**Table 1.** Comparison of relevant open-sourced high level control systems. Source [3].

| Flight Stack | Open Source | Modular | Tested in | Middle-Ware | Soft. Last Update | Multi-Agent | Multi-Platform |
|---|---|---|---|---|---|---|---|
| Aerostack2 [3] | ✓ | ✓ | S,RL,RO | ROS 2 | 03/2023 | ✓ | ✓ |
| Aerostack [12] | ✓ | ✓ | S,RL,RO | ROS | 10/2021 | ✓ | ✓ |
| AerialCore [13] | ✓ | ✓ | S,RL,RO | ROS | 03/2023 | ✓ | ✗ |
| Agilicius [14] | ✓ | ✓ | S,RL | ROS | 03/2023 | ✗ | ✗ |
| CrazyChoir [15] | ✓ | ✗ | S,RL | ROS 2 | 02/2023 | ✓ | ✗ |
| KumarRobotics [16] | ✓ | ✗ | S,RL,RO | ROS | 12/2022 | ✗ | ✓ |
| UAL [17] | ✓ | ✗ | S,RL,RO | ROS | 12/2022 | ✗ | ✓ |

S: simulation, RL: real experiments in the lab, RO: real experiments outside the lab.

This paper presents two use cases related to autonomous industrial inspections, which have been addressed in various works within the field.

Regarding the first proposed use case, which involves the inspection of a wind turbine, ref. [18] describes how the inspection of the system is carried out while it is in operation, which is controlled by a pilot operating the UAV. However, when the aim is to automate the process, as proposed in [19], the wind turbine is stopped, and the inspection of the static blades is performed. In the mentioned work, an innovative autonomous

inspection of the system while in operation is presented, which implies a higher level of safety for the developed algorithms.

As for the second case, focusing on the inspection of a photovoltaic plant, there are works such as [20] that propose conducting the inspection through manual flight, while others like [21] propose automation. Nevertheless, the latter have not been able to implement it in real flights due to the risks and costs involved. Therefore, the autonomous inspection presented in this article, which also incorporates the multi-agent feature, required an efficient and secure development methodology to achieve successful realization.

## 3. Proposed Methodology

This work proposes a methodology for the development of robotic applications, aiming to optimize development time and ensure security [22]. Specifically, in the context of industrial inspection applications using aerial robots, it is crucial to minimize the test time during real flights due to the associated costs. Moreover, the robustness of the system must be guaranteed to prevent potential damage to the facilities.

The development of autonomous aerial robotics systems typically involves two stages [23–25]. The first stage entails simulating both the aircraft and its environment as accurately as possible. Subsequently, in the second stage, real flights are conducted within industrial facilities.

To achieve a more efficient and secure development process, this work introduces a novel methodology that incorporates additional stages, namely:

1. Simulation-based development.
2. Hardware-integration validation.
3. Augmented-reality validation.
4. Industrial environment validation.

Table 2 details the status of each component in the different phases of the proposed methodology process. In Figure 1, these components are graphically depicted.

**Table 2.** Components of the methodology at each stage of the development process.

| Stage | Aerial Platform | Hardware | Flight | Industrial Facility |
|---|---|---|---|---|
| SbD | Simulator | SITL | Simulated | Simulated |
| HiV | Real Aircraft | HIL | Simulated | Simulated |
| AR-V | Real Aircraft | Real | Real | Simulated |
| IEV | Real Aircraft | Real | Real | Real |

SbD: Simulation-Based Development, HiV: Hardware-Integration Validation, AR-V: Augmented-Reality Validation, IEV: Industrial Environment Validation.

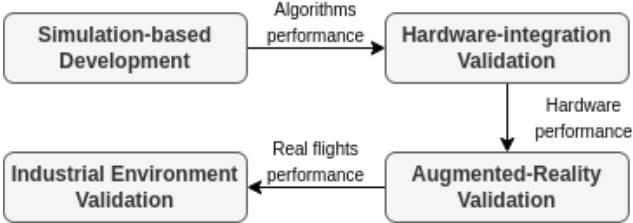

**Figure 1.** Each of the stages of the methodology, where the connections between them indicate the components that have been validated to progress to the next stage.

### 3.1. Simulation-Based Development

The first stage of development involves simulating the aircraft, environment, and algorithms within a simulation station. This stage serves as a crucial foundation for the development process [26], allowing engineers to design, test, and refine the system's components in a controlled and virtual environment. In the simulation station, engineers

create a realistic virtual representation of the aircraft and its surrounding environment, including the physical properties, dynamics, and potential obstacles. They also incorporate the algorithms and control mechanisms that govern the system's behavior.

Simulating in the simulation station provides several benefits [2]. Firstly, it allows engineers to validate and fine-tune the algorithms and control strategies before implementing them in the physical system. They can analyze the system's performance under different scenarios and assess its responsiveness to various inputs and environmental conditions. By conducting extensive simulations, engineers can optimize the system's behavior, improve its efficiency, and enhance its overall performance.

Secondly, the simulation station enables engineers to reduce the risk and costs associated with real-world testing. Instead of relying solely on physical prototypes and actual flight tests, engineers can iterate through numerous design iterations and scenario simulations. They can test the system's capabilities in challenging or hazardous conditions without endangering personnel or equipment. This significantly accelerates the development process and minimizes the potential for costly errors or accidents during the early stages.

In this stage, engineers can also employ software-in-the-loop (SITL) simulations. SITL serves as an additional simulator within the simulation station, encompassing the flight software and enabling the transmission and reception of flight commands without the presence of an actual controller [26]. Typically, it supports the simulation of basic sensors like GPS and IMU. Integrating SITL into the simulation process allows engineers to further refine and evaluate the behavior of the system, ensuring its compatibility with the selected software and controller configuration.

### 3.2. Hardware-Integration Validation

The second stage of development involves simulating the environment in the simulation station: processing algorithms are executed on the on-board computer that we used in the actual flight, and we are simulating the platform using the hardware-in-the-loop (HIL) technique. HIL is a technique that is used in the development and testing of complex real-time embedded systems like a drone [27]. In this stage, engineers integrate the simulated environment with the actual on-board computer and drone hardware, creating a hardware and software co-simulation setup. The on-board computer runs the algorithms and interacts with the simulated environment in real time, providing a comprehensive, cost-effective, and repeatable testing manner.

Simulating in the deployment on-board computer offers several advantages [27]. Firstly, it allows engineers to assess the performance of the algorithms and on-board computer under realistic hardware limitations. By testing the system on the actual hardware, engineers can identify any potential computational constraints, memory limitations, or processing delays that may affect the system's overall performance. This stage helps ensure that the algorithms and on-board computer are optimized and capable of meeting the system's requirements in real-world scenarios.

The HIL simulation enables engineers to evaluate the platform's response to the algorithms processed by the on-board computer. Simulating in the deployment hardware facilitates the integration of various components and subsystems, ensuring their seamless interaction and compatibility. Engineers can verify the communication protocols, sensor interfaces, and data-exchange mechanisms, ensuring smooth operation and coordination among the system's different elements. By addressing any integration challenges early on, engineers can streamline the subsequent stages of development and enhance the system's overall reliability and performance.

### 3.3. Augmented-Reality Validation

The third stage of development is a crucial step that involves transitioning from hardware-in-the-loop (HIL) simulation to real-world testing through actual flight. In this stage, the aircraft is equipped with the on-board computer and all necessary components,

enabling comprehensive evaluation of its performance and functionality in real-world conditions. This critical phase aims to validate the system's readiness for practical deployment by assessing its ability to execute intended tasks accurately, efficiently, and reliably in the real world.

Additionally, in order to enhance the safety and efficiency of the real-world testing process, industrial facilities are simulated simultaneously with the drone flights. By incorporating simulation data into the testing phase, engineers can avoid flying the drone directly over or near industrial facilities, reducing the risk of potential accidents or damage. This approach allows for a comprehensive evaluation of the drone's flight performance while ensuring that it operates within predefined safety boundaries. The simulated industrial facilities provide realistic environmental factors, such as obstacles, structures, and varying conditions, enabling engineers to assess the drone's ability to navigate and respond effectively in complex operational settings. By combining augmented-reality validation with simulated industrial environments, the testing process becomes more robust, enabling thorough assessment and refinement of the drone's capabilities before practical deployment.

When replacing the HIL simulation with an actual flight, engineers obtain invaluable insights into the system's behavior and responsiveness in real-world scenarios. It provides an opportunity to evaluate how the system performs when subjected to the dynamic and unpredictable nature of the operational environment. This stage plays a vital role in verifying that the system's design and functionality align with the intended goals and requirements.

During the real-world testing, engineers closely monitor the aircraft's performance, flight dynamics, and control responsiveness. They assess its capability to execute planned maneuvers, navigate through various scenarios, and adapt to changing conditions encountered during the flight. By observing the system's behavior in real time, engineers can identify any discrepancies, anomalies, or performance limitations that may arise.

*3.4. Industrial Environment Validation*

The final stage of development involves testing the system in actual industrial installations where it will be deployed. The goal is to evaluate its performance and functionality in a real-world operational setting, considering the specific conditions and challenges of the deployment environment.

During this stage, engineers conduct thorough tests to validate the system's effectiveness and efficiency in performing the intended tasks. They assess factors like navigation, object detection and recognition, data processing speed, and system responsiveness. Testing in the deployment environment ensures reliable and accurate operation in real-world scenarios.

Engineers also identify potential issues or limitations due to the unique characteristics of the industrial installations. They evaluate the system's adaptability to lighting conditions, environmental factors, and facility obstacles. Additionally, they assess the system's robustness and resilience in handling unexpected situations or failures, ensuring safe and effective operation.

Stakeholders and end-users participate in the testing process, providing valuable input on usability, user interface, and satisfaction. Incorporating user perspectives refines the system's design and functionality to meet the specific requirements and expectations of the deployment environment.

The collected data are analyzed and compared with expected performance benchmarks. Discrepancies are investigated and addressed to enhance system performance and reliability. Iterative testing and refinement ensure continuous improvement and optimization of the system's capabilities.

Completing this final stage of testing in the deployment environment enables confident deployment of the system for operational use. The comprehensive evaluation provides assurance of effective, safe, and efficient task performance within the industrial setting. It

also allows engineers to gather insights and feedback for further enhancements and fine-tuning. Ultimately, testing in the deployment environment ensures the robotic application is fully prepared to meet operational requirements and deliver desired outcomes in the industrial context.

## 4. Implementation

In this section, the implementation of the proposed methodology is explained, analyzing how the transition from simulation to the real world has been carried out.

Aerostack2 [3] is specialized in aerial robotics and is applicable to different types of platforms, both real (e.g., DJI Matrice, Crazyflie, etc.) and simulated (e.g., using the Gazebo simulator). Built on the foundation of ROS 2 (Robot Operating System 2) [28], an open-source middleware specifically designed for robotic systems, Aerostack2 enables a seamless integration of diverse software components. This facilitates the construction of intricate and tailored robotic systems while ensuring real-time communication among their various components. Notably, ROS 2's distributed architecture plays a key role, as it allows information to be distributed across a network accessible by different execution nodes. This capability empowers the creation of highly complex robotic systems.

A noteworthy aspect of the Aerostack2 framework is its interaction with the Aerial Platform interface. This interaction facilitates the integration of both physical and simulated interfaces without requiring the rest of the framework to differentiate between them. This platform-agnostic feature serves as a crucial pillar, strengthening the capabilities of the developed algorithms in transitioning from simulation to the real-world environment [3].

To facilitate simulation, Aerostack2 offers a platform based on the Gazebo simulator [8], which is a physics simulator. Moreover, compatibility with Unity, a graphics engine, is achieved through the utilization of a customized version of Flightmare [11] for ROS 2.

### 4.1. Gazebo Simulator Integration

The Aerostack2 Gazebo platform is responsible for relaying control commands to the simulated UAV. They are received from the motion controller and transmitted through the platform. As the platform is using a common interface, it can be easily decoupled for a real UAV's platform, which will receive the control commands to be sent to the UAV exactly the same way.

This platform utilizes the ROS 2–Gazebo bridge library (https://github.com/gazebosim/ros_gz, accessed on 23 June 2023), which acts as a bridge between ROS 2 and Gazebo, facilitating two-way communication for simulating and testing robots in a virtual environment. It enables the creation of an interface between Gazebo's communication mechanism and the one used in ROS 2. With this platform, users can control and monitor the behavior of simulated robots in Gazebo through Aerostack2.

To define the simulation setup, a configuration file is used to define aerial robots and external objects. This file is then loaded by an Aerostack2 component responsible for launching the simulation. This component includes all the assets of each model specified in the configuration file as well as any ROS 2 to Gazebo bridges unrelated to aerial robots. For the aerial robots and their sensors defined in the configuration file, the Aerostack2 Gazebo platform handles the launching of the corresponding ROS 2 to Gazebo bridges. The overall scheme is illustrated in Figure 2a.

After this, through the ROS 2–Gazebo bridge, Aerostack2 can communicate with the simulator using the ROS 2 common communication mechanisms: these include topics, services, actions, and transformation frames (tf2), as shown in Figure 2b.

### 4.2. Simulation of the UAV

The UAV simulation in Gazebo primarily relies on the *Multicopter Velocity Control* plugin (https://gazebosim.org/api/gazebo/4.3/classignition_1_1gazebo_1_1systems_1_1MulticopterVelocityControl.html, accessed on 23 June 2023), which enables control of the linear velocity and yaw angular velocity of the vehicle. This plugin requires a quadrotor

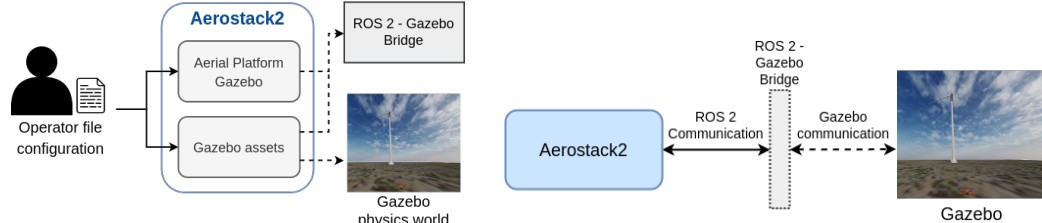

(**a**) Gazebo simulator launch scheme.      (**b**) Gazebo-Aerostack2 communication.

**Figure 2.** Gazebo simulator launch scheme in Aerostack2 and its communication with the framework.

with a minimum of four rotors (*Multicopter Motor Model* plugin) to function. Additionally, other Gazebo plugins, such as a magnetometer, an IMU, and a GPS receiver, are included in the quadrotor simulation.

In the context of hardware-in-the-loop (HIL) simulation, Aerostack2 utilizes the DJI Assistant 2 software (https://www.dji.com/global/downloads/softwares/assistant-dji-2, accessed on 23 June 2023), which allows the simulation of DJI drones without modifying the platform interface.

To facilitate this integration, the Aerostack2 framework adopts a standardized structure, as depicted in Figure 3, where the interface with various drones, both simulated and real, is consolidated under a unified node known as the Aerial Platform.

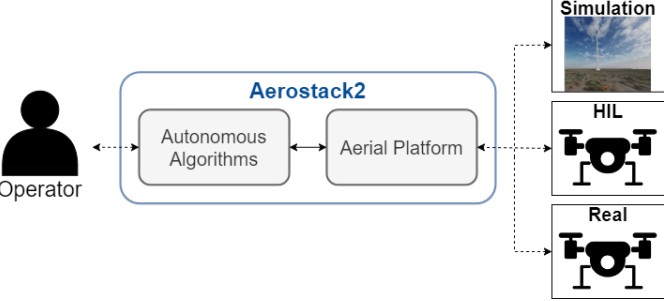

**Figure 3.** Simulation scheme in Aerostack2, where the aircraft can be interchangeable, keeping the same algorithms and only modifying the aerial platform.

## 5. Use of Case for Autonomous Wind Turbine Inspection

In the following sections, the proposed methodology will be validated in two relevant uses of cases. The first scenario consists of the inspection of a wind turbine during its regular operation. This case study is particularly dangerous, since both the blades and nacelle are rotating. The nacelle is connected to the rotor, which makes the blades spin, and also to the tower, which is rotating along its z-axis.

The mission consists of following several inspection points linked to the wind turbine rotor. These points are calculated depending on certain inputs such as the blade's length, the intrinsic camera parameters, or the security inspection distance.

The prior data received from the wind turbine include the position in WGS84 coordinates of the base of the wind turbine, the blades' length, and the rotor height. The real-time data received from the wind turbine include the orientation of the rotor given in azimuth (horizontal angle from north).

### 5.1. Simulation of the Wind Turbine

For the simulation of the wind turbine, a model of a wind turbine has been created in the Blender [29] tool. This model has been divided into three main parts. These parts are then turned into separate SDFormat (http://sdformat.org/spec?ver=1.9&elem=sdf, accessed on 23 June 2023) models and connected by rotary joints. These parts are outlined below:

- Tower: This is the static part of the wind turbine linked to the ground. The base of the tower has been set as the origin of our wind turbine coordinate system.
- Nacelle: It consists of a box-like structure that connects the tower with the rotor. The top of the tower is connected by a rotation joint that rotates in yaw (X-Y plane).
- Blades: This model contains the rotor with the blades. The rotor is connected to the front of the nacelle with another rotation joint that rotates in roll (Y-Z plane).

In order to move the blades and the nacelle within the simulation, Gazebo's joint speed controller plugins have been used for each of the joints within the model. These plugins have then been bridged to ROS 2.

A simulated GPS sensor has been integrated into the nacelle so we can receive the WGS84 coordinates. The cartesian orientation of this sensor is used to calculate the azimuth, which is then sent through an ROS 2 topic. Its primordial function is to obtain real-like data structures and values from the simulation.

As the mission is planned with poses relative to the wind turbine rotors, the nacelle GPS coordinate and the azimuth have to be transformed into cartesian positions. When the information arrives to the UAS, it is transformed into a pose and then added to the transformation tree relative to the global reference frame. This way, we can address the problem of navigating in the global coordinate system with the given relative poses.

*5.2. Deployment*

The methodology used to address this problem consists of using a fully simulated environment first and then starting to substitute simulated components for real components gradually in order to finally deploy the mission in a real environment in a secure and efficient way. This process has taken three steps before deploying every component in a real environment.

5.2.1. Simulation-Based Development

Every hardware component is simulated in the Gazebo simulator. Aerostack2 receives the data bridged from Gazebo coming from the UAV and the wind turbine. The control commands for both the wind turbine and the UAV are bridged from ROS 2 and sent to Gazebo, as shown in Figure 4a.

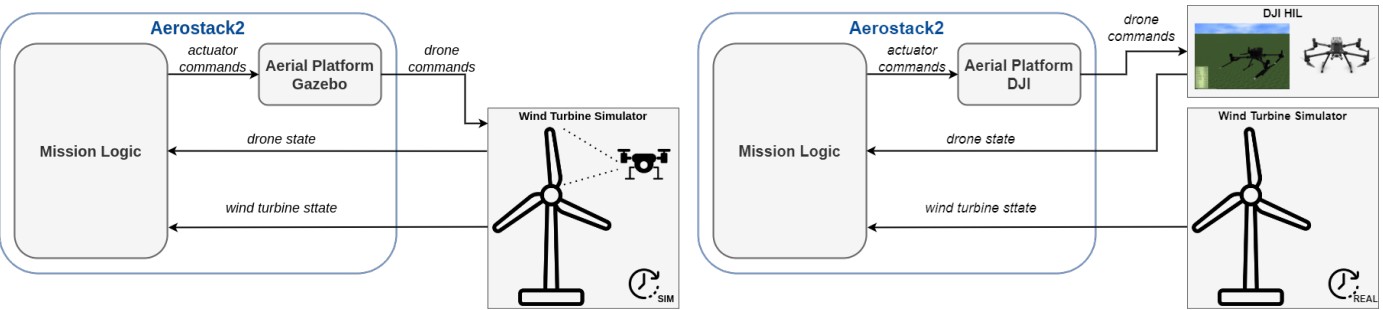

(**a**) Simulation-based development scheme.                    (**b**) Hardware-integration validation scheme.

**Figure 4.** *Cont.*

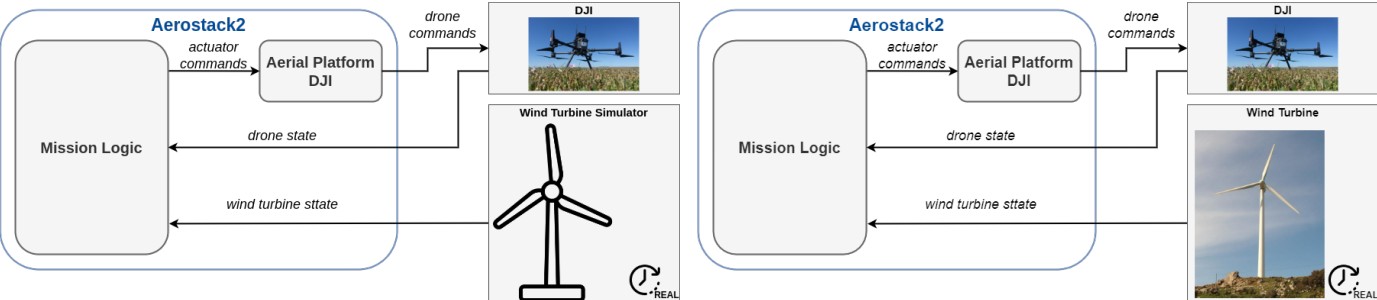

(**c**) Augmented-reality validation scheme.  (**d**) Industrial environment validation.

**Figure 4.** Scheme of the system through the different development stages.

### 5.2.2. Hardware-Integration Validation

Gazebo's basic quadrotor is detached from the simulation and substituted for DJI's hardware in the loop (HIL), as seen in Figure 4b. This step ensures that the platform that is going to be used in the real flight works with the initial mission planning and data regarding the wind turbine.

### 5.2.3. Augmented-Reality Validation

The last step before real deployment consists of substituting DJI's HIL for the real hardware, as shown in Figure 4c. The wind turbine simulation allows testing the aerial system in real flights without endangering a wind turbine.

In order to monitor the mission planning and execution, all the information coming from every component used in this step has been integrated into the RViz2 [30] visualizer. The information integrated can be seen in Figure 5a.

In Figure 5b, the actual trajectory followed by the drone during the inspection is depicted, showcasing its adaptation to the rotation of the nacelle as the inspection progresses. As the drone conducts the inspection, it dynamically adjusts its trajectory to accommodate the changing orientation of the nacelle. This adaptive behavior is facilitated by receiving real-time position and orientation information from the simulator.

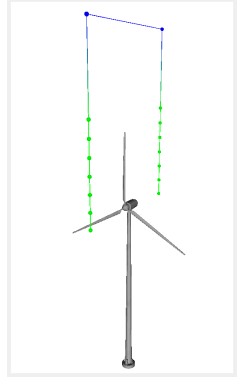  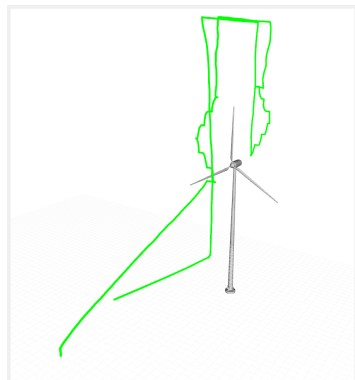

(**a**) Inspection waypoints.  (**b**) Trajectory carried out by the drone.

**Figure 5.** Planned trajectory for the inspection of the wind turbine (**a**) and the one followed by the autonomous drone adapting to the rotation of the nacelle (**b**) in the real flight.

### 5.2.4. Industrial Environment Validation

The final stage of the deployment process involves the crucial validation of the system in the industrial facilities. This step focuses on conducting thorough inspections of the wind turbine while integrating all the necessary systems to ensure seamless operations.

One integral component is the ground station, which is responsible for computing the precise position and orientation of the nacelle. This information is then transmitted to the drone, enabling it to navigate accurately and carry out its inspection tasks efficiently.

## 6. Use of Case for Autonomous Photovoltaic Plant Inspection

This particular application involves the examination of operational photovoltaic plant panels. The mission consists of covering the inspection area related to the lines of photovoltaic panels with multiple UAVs simultaneously. This entails capturing images using both color and thermal cameras for subsequent analysis and defect detection.

This scenario, in which a swarm of drones is tasked with conducting inspections above expensive industrial facilities, presents significant safety concerns. Managing multiple UAVs in real-world experiments presents significant challenges. Our methodology aims to address this problem by reducing the complexity of such application.

To facilitate this process, a georeferenced map of the photovoltaic plant serves as the foundational data. This map allows for the definition of the inspection path within the plant.

### 6.1. Simulation of Photovoltaic Plant

To simulate a photovoltaic plant, a 2D georeferenced model of the plant has been developed in Gazebo, shown in Figure 6a, serving as the background floor. Additionally, a photorealistic world has been designed in the Unity simulator for image capture purposes, as depicted in Figure 6b. To establish a connection with ROS 2, Flightmare [11] has been utilized.

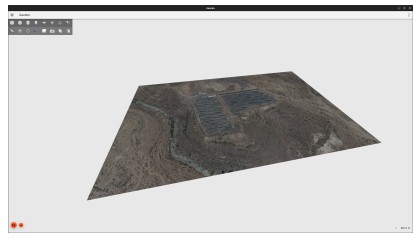
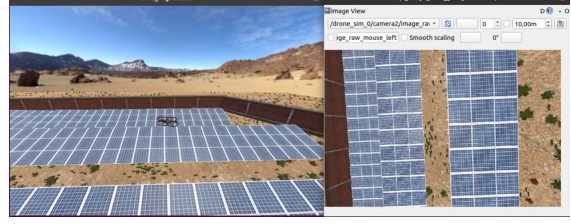

(**a**) Simulation in Gazebo.     (**b**) Unity simulation and drone's camera view.

**Figure 6.** Simulation of the photovoltaic plant used during the development process.

To facilitate mission planning, the Aerostack2 Web–GUI ground station, shown in Figure 7, has been utilized. This user interface enables the definition of optimal paths for each UAV involved in the inspection process. The planned paths are designed to ensure that each image captured satisfies the given inspection parameters.

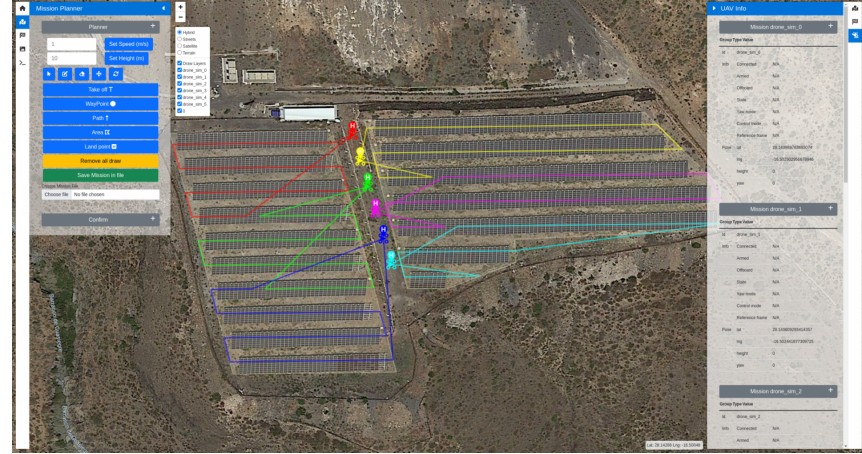

**Figure 7.** Aerostack2 Web–GUI used to plan the photovoltaic plant inspection, where the inspection paths of each UAV are displayed in different colors.

## 6.2. Deployment

As previously stated, the proposed methodology involves a progressive substitution of simulated components with real components. In this particular scenario, the simulation of the swarm is replaced with actual drones, while the simulated data from the photovoltaic plant are utilized, as depicted in Figure 8.

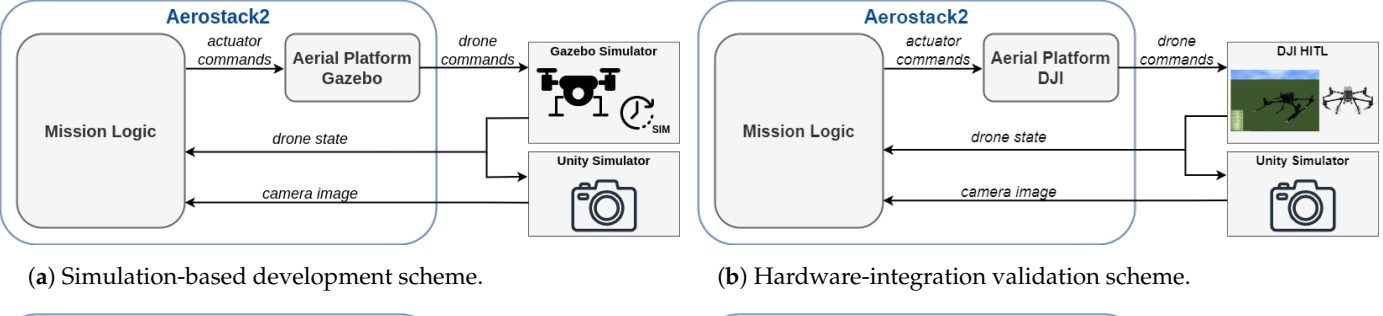

(**a**) Simulation-based development scheme.      (**b**) Hardware-integration validation scheme.

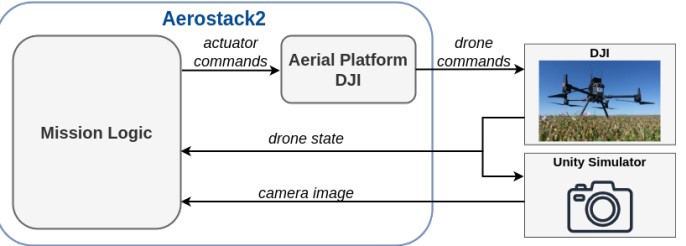 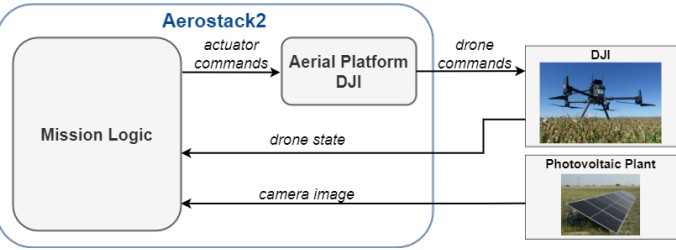

(**c**) Augmented-reality validation scheme.      (**d**) Industrial environment validation.

**Figure 8.** Scheme of the system through the different development stages.

### 6.2.1. Simulation-Based Development

The UAV swarm is simulated using Gazebo, while the camera simulation takes place in Unity. Aerostack2 receives data bridged from Gazebo, which originate from the UAV, as well as the image generated by Unity. The control commands for the UAV are bridged from ROS 2 and transmitted to Gazebo. Additionally, the drone's pose extracted from Gazebo is sent to both Aerostack2 and Unity. Refer to Figure 8a for a visual representation of this process.

### 6.2.2. Hardware-Integration Validation

In the subsequent step, the standard quadrotor model in Gazebo is removed from the simulation and replaced with DJI's hardware in the loop (HIL) system, as depicted in Figure 8b. This stage guarantees compatibility between the actual flight platform and the mission logic algorithms, while the image data continue to be sourced from Unity.

### 6.2.3. Augmented-Reality Validation

In the final phase prior to actual deployment, DJI's HIL system is replaced with the real hardware, as illustrated in Figure 8c. By simulating the photovoltaic plant in Unity, the aerial system can undergo real flight testing without posing any risks to the solar panels.

### 6.2.4. Industrial Environment Validation

The last step in the deployment process focuses on the validation of the system within a photovoltaic plant. Figure 9a showcases the trajectories followed by a swarm of two drones as they inspect the designated area within the plant. During the inspection, the drones navigate in a coordinated manner, following predetermined trajectories to ensure comprehensive coverage of the photovoltaic panels.

Figure 9b showcases an image captured from the ground perspective, where one of the drones can be seen taking photos of the photovoltaic panels. The image provides a

visual representation of the inspection process, highlighting the drone's role in capturing detailed images of the panels from an aerial viewpoint.

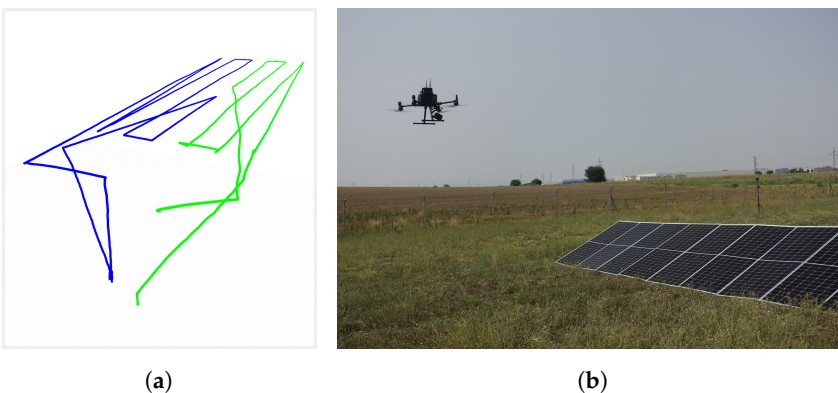

| (**a**) | (**b**) |
|---|---|

**Figure 9.** Real trajectory followed by two autonomous drones during the area inspection of a photovoltaic plant (**a**) and a photograph captured during the inspection, showing one of them capturing images of the photovoltaic panels (**b**).

## 7. Experimental Evaluation

In this section, the results obtained from the execution of the proposed methodology for each of the presented industrial applications will be presented.

### 7.1. Wind Turbine Inspection

In this experiment, the inspection of a wind turbine has been carried out both on the front and rear parts of the blades. Thus, from the inertial reference point of the nacelle, the drone must follow a vertical trajectory, where it captures images at different heights. From an external reference point, the drone's trajectory will be vertical with rotations relative to the nacelle, adapting to its rotation, as shown in Figure 5b.

For the first stage based on simulation, as explained in Section 5.2.1, the Gazebo simulator has been used. For both hardware-in-the-loop (HIL) and flight with augmented reality, a DJI Matrice 300 aircraft has been used with an NVIDIA Jetson AGX Xavier computer on board.

Figure 10 shows the results obtained in each of the methodology stages. Table 3 presents the trajectory tracking errors obtained, displaying distance errors in the first part and orientation errors in the second part. These metrics are essential to ensure a proper wind turbine inspection.

**Table 3.** Results of wind turbine inspection. Presented are the errors between the intended trajectory and the executed trajectory.

| (a) Distance errors. | | | |
|---|---|---|---|
| **Stage** | **Mean Error (m)** | **Max Error (m)** | **Error SD (m)** |
| SbD | 0.11 | 0.31 | 0.06 |
| HiV | 0.11 | 0.22 | 0.04 |
| AR-V | 0.13 | 0.41 | 0.08 |

| (b) Orientation errors. | | | |
|---|---|---|---|
| **Stage** | **Mean Error (rad)** | **Max Error (rad)** | **Error SD (rad)** |
| SbD | 0.02 | 0.10 | 0.01 |
| HiV | 0.05 | 0.10 | 0.04 |
| AR-V | 0.02 | 0.08 | 0.01 |

SbD: Simulation-Based Development, HiV: Hardware-Integration Validation, AR-V: Augmented-Reality Validation.

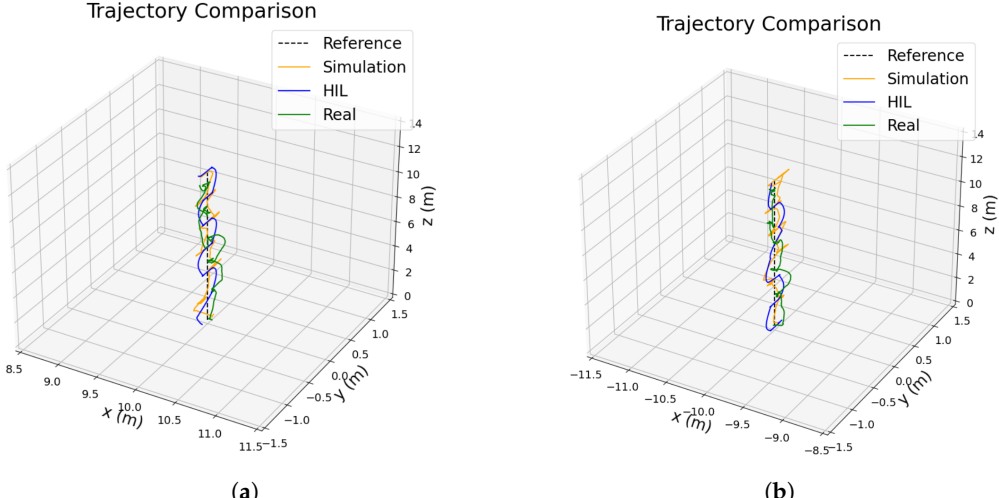

(**a**)                                                    (**b**)

**Figure 10.** Trajectory followed by the autonomous drone during the inspection of a wind turbine, encompassing both the frontal aspect of the blades (**a**) and the rear section (**b**). The figure illustrates the reference trajectory from the nacelle inertial frame (depicted in black and dashed lines), the simulated trajectory (illustrated in orange), the hardware-validated trajectory (displayed in blue), and the trajectory recorded during live flight (shown in green).

*7.2. Photovoltaic Plant Inspection*

In this second experiment, the inspection of a photovoltaic plant with five lines of panels has been conducted. For this purpose, two aircraft have been used, flying at a maximum speed of 2 m/s, which are capable of simultaneously inspecting the desired area.

For the first stage based on simulation, in the same way as the previous experiment, the Gazebo simulator has been used. Regarding the hardware, both aircraft have been equipped with an NVIDIA Jetson AGX Xavier computer onboard, with one drone being a DJI Matrice 200 and the other being a DJI Matrice 300.

In Figure 11, the trajectories covered by each drone in each stage of the methodology are shown. Tables 4 and 5 display the errors incurred in trajectory tracking, showing distance errors in the first table and orientation errors in the second.

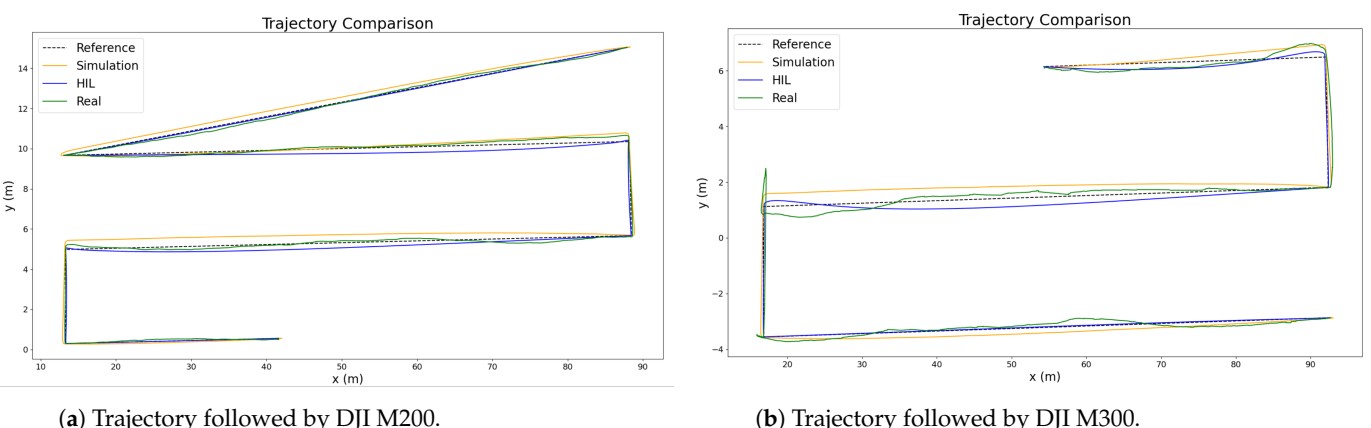

(**a**) Trajectory followed by DJI M200.                    (**b**) Trajectory followed by DJI M300.

**Figure 11.** Trajectory pursued by autonomous drones during the inspection of a photovoltaic plant, employing two different models, DJI M200 (**a**) and DJI M300 (**b**), simultaneously. The figure depicts the reference trajectory for the area inspection (represented by black dashed lines), the simulated trajectory (illustrated in orange), the hardware-validated trajectory (displayed in blue), and the trajectory recorded during live flight (shown in green).

**Table 4.** Results of photovoltaic plant inspection by two drones simultaneously. Presented are the distance errors between the intended trajectory and the executed trajectory.

**(a) Results of DJI M200.**

| Stage | Mean Error (m) | Max Error (m) | Error SD (m) | Time (s) |
|-------|----------------|---------------|--------------|----------|
| SbD   | 0.20           | 0.45          | 0.14         | 155.84   |
| HiV   | 0.12           | 0.29          | 0.10         | 131.86   |
| AR-V  | 0.10           | 0.31          | 0.07         | 135.99   |

**(b) Results of DJI M300.**

| Stage | Mean Error (m) | Max Error (m) | Error SD (m) | Time (s) |
|-------|----------------|---------------|--------------|----------|
| SbD   | 0.23           | 0.47          | 0.15         | 115.93   |
| HiV   | 0.11           | 0.31          | 0.10         | 103.02   |
| AR-V  | 0.15           | 0.65          | 0.12         | 124.60   |

SbD: Simulation-Based Development, HiV: Hardware-Integration Validation, AR-V: Augmented-Reality Validation.

**Table 5.** Results of photovoltaic plant inspection by two drones simultaneously. Presented are the orientation errors between the intended trajectory and the executed trajectory.

**(a) Results of DJI M200.**

| Stage | Mean Error (rad) | Max Error (rad) | Error SD (rad) |
|-------|------------------|-----------------|----------------|
| SbD   | 0.00             | 0.01            | 0.00           |
| HiV   | 0.00             | 0.05            | 0.01           |
| AR-V  | 0.07             | 0.12            | 0.01           |

**(b) Results of DJI M300.**

| Stage | Mean Error (rad) | Max Error (rad) | Error SD (rad) |
|-------|------------------|-----------------|----------------|
| SbD   | 0.00             | 0.01            | 0.00           |
| HiV   | 0.00             | 0.03            | 0.00           |
| AR-V  | 0.28             | 0.40            | 0.01           |

SbD: Simulation-Based Development, HiV: Hardware-Integration Validation, AR-V: Augmented-Reality Validation.

## 8. Conclusions and Future Work

In this research, we have emphasized the necessity of developing industrial applications utilizing unmanned aerial systems in an efficient and safe manner. We have highlighted the convenience of using an incremental sim2real strategy for isolating the possible causes of failures during the development stages. The proposed methodology successfully meets these requirements and has been applied to two specific use cases: wind turbine inspection and photovoltaic plant inspection.

In both conducted experiments, similar performances have been achieved in each stage of the methodology, confirming its applicability. This is because performance improvements and error resolution in one stage lead to corresponding enhancements in the subsequent stages.

A noteworthy contribution of our work is the introduction of a novel stage of mixed reality, serving as an intermediary step to conduct real flights without endangering industrial facilities.

The integration of the Aerostack2 framework has proven advantageous in this regard, facilitating seamless progression through the various stages with minimal changes to system components. Aerostack2 plays a pivotal role in bridging the gap between simulation and reality, enhancing the overall effectiveness of the methodology.

Looking ahead, there are several avenues for further research and improvement. One key area of focus involves enhancing the realism of simulations to minimize discrepancies

between simulated and real installations. This refinement would significantly contribute to more accurate validation processes at each stage.

Furthermore, the proposed methodology is generic for any industrial application, so a future task is to apply it to new applications and assess its performance.

Additionally, we propose exploring the utilization of new quantitative tools to support the validation of individual components and stages within the methodology. These advancements will enable safer and more efficient operations, minimizing risks to the system, industrial facilities, and personnel involved.

**Author Contributions:** Conceptualization, R.P.-S., P.A.-P., M.F.-C. and P.C.; methodology, R.P.-S. and P.A.-P.; software, R.P.-S., P.A.-P., J.M.-D., M.F.-C. and D.P.-S.; validation, R.P.-S., P.A.-P., J.M.-D., M.F.-C. and D.P.-S.; formal analysis, R.P.-S. and P.A.-P.; investigation, R.P.-S., P.A.-P. and J.M.-D.; resources, P.C.; data curation, R.P.-S.; writing—original draft preparation, R.P.-S.; writing—review and editing, R.P.-S., P.A.-P., M.F.-C., J.M.-D. and P.C.; supervision, P.C.; project administration, P.C.; funding acquisition, P.C. All authors have read and agreed to the published version of the manuscript.

**Funding:** This work has been supported by the project COPILOT ref. Y2020\EMT6368 "Control, Monitoring and Operation of Photovoltaic Solar Power Plants by means of synergic integration of Drones, IoT, and advanced communication technologies", funded by Madrid Government under the R&D Synergic Projects Program. We acknowledge the support of the European Union through the Horizon Europe Project No. 101070254 CORESENSE. This work has also been supported by the project INSERTION ref. ID2021-127648OBC32, "UAV Perception, Control and Operation in Harsh Environments", which was funded by the Spanish Ministry of Science and Innovation under the program "Projects for Knowledge Generating". In addition, this work has also been supported by the project RATEC ref: PDC2022-133643-C22 "Localization and planning of thetered aerial+ground robots for inspection and maintenance tasks" funded by the Spanish Ministry of Science and Innovation. The work of the second author is supported by the Grant FPU20/07198 of the Spanish Ministry for Universities. The work of the fifth author is supported by the Spanish Ministry of Science and Innovation under its Program for Technical Assistants PTA2021-020671.

**Data Availability Statement:** Not applicable.

**Acknowledgments:** The authors would like to thank Aeromedia UAV S.L. for providing the wind turbine inspection specifications and Avistadrone S.L. for providing the flight area necessary to carry out the experiments.

**Conflicts of Interest:** The authors declare no conflict of interest.

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
