# Peer review of "Bridging the Gap between Simulation and Real Autonomous UAV Flights in Industrial Applications"

_aerospace, doi:10.3390/aerospace10090814_

Round 1

Reviewer 1 Report

The aim of the paper is the introduction of intermadiate stages between simulation and real life deployment of UAV. The advantage of the proposed methodology is obvious.

The article is mostly very well presented. However, I miss images in each of chapters 3.1 to 3.4 which would clearly depict each stage and the difference between them.

Additionally, I think the results are very scarce in the article. I would expect to see more results as there were actual tests performed. There are no numbers, just couple of images.

Also, were there any unexpected offsets between simulations and real life applications? If not, where could they appear?

Conclusions could be backed up with the results.

I have no specific remarks/questions, just general.

Author Response

We would like to express our gratitude for your insightful review of our manuscript. Your feedback has been immensely valuable in refining our work. We are pleased to inform you that we have taken your suggestions into consideration and made the necessary revisions to improve the overall quality of the paper. We are attaching the revised manuscript with the modifications implemented based on the feedback of both reviewers, which have been highlighted in blue. Thank you for your time and consideration.

“The article is mostly very well presented. However, I miss images in each of chapters 3.1 to 3.4 which would clearly depict each stage and the difference between them.”

We consider that having a diagram in each section is challenging, as it depends on the implementation. Therefore, we have decided to leave those diagrams in the later chapters. However, we have added Figure 1 to provide a visual aid and to enhance the understanding of that specific chapter.

“Additionally, I think the results are very scarce in the article. I would expect to see more results as there were actual tests performed. There are no numbers, just couple of images.”

“Also, were there any unexpected offsets between simulations and real life applications? If not, where could they appear?”

“Conclusions could be backed up with the results.”

We have included an experiments chapter. Within it, we present certain metrics that enable the validation of the proposed methodology's applicability. Furthermore, we assess the offsets between simulation and reality. However, as stated in the conclusions, future work will focus on minimizing this disparity. It's important to note that, given that this isn't a control article, the significance of the presented data lies in the resemblance between various developmental stages rather than the absolute values themselves.

Reviewer 2 Report

This manuscript is very interesting and relevant, however suffers from many basic weakness. My main problem is that the manuscript sounds like a project report and not like a scientific paper. Against my critics I motivate and suggest authors, to make basic revision and summit it again.

The title suggests that general conclusions can be drawn from the results, however, based on the two practical examples, this cannot yet be done. I recommend that the authors clarify and modify the title. 

The abstract does not follow IMRaD requirements, so I recommend the authors to revise this section significantly accordingly.

I recommend putting the keywords in alphabetical order.

The Introduction lacks the broader context of the topic. This chapter contains only one reference! Although the formulations are acceptable, they are very general. I recommend that the authors place their work in a broader context, justify the need to investigate the topic and provide references.

Right at the beginning, I note that the manuscript contains only 23 references, which is a clear sign that the manuscript is not sufficiently processed from a scientific point of view and does not meet the basic requirements for scientific works.

The Introduction does not state what the specific objective is. Please, specify it accurately.

I recommend that the contributions be reworded and objectives given in the introduction.

I recommend that the structure of the manuscript be formulated at the end of the Introduction.

 Detail of Related work is acceptable.

 The Methodology chapter is completely lacking in contextualization, it does not contain a single reference!

A summary of the specific results is missing from the end of the manuscript. The wording of the Conclusions chapter is excessively general. Please, reconstruct, complement and specify it.

Overall, the manuscript contains important and relevant information and research, but in its current form it is too poor to be published as a scientific publication in this journal.

I encourage authors to revise thoroughly and resubmit a modified version.

Acceptable, only minor spelling is required.

Author Response

We would like to express our gratitude for your insightful review of our manuscript. Your feedback has been immensely valuable in refining our work. We are pleased to inform you that we have taken your suggestions into consideration and made the necessary revisions to improve the overall quality of the paper. We are attaching the revised manuscript with the modifications implemented based on the feedback of both reviewers, which have been highlighted in blue. Thank you for your time and consideration.

“The title suggests that general conclusions can be drawn from the results, however, based on the two practical examples, this cannot yet be done. I recommend that the authors clarify and modify the title”

We have kept the title as we believe that with the added changes, it aligns better with the article. However, we do have other more specific alternatives.

“The abstract does not follow IMRaD requirements, so I recommend the authors to revise this section significantly accordingly.”

Likewise, we believe that the presented abstract fits well with the made changes and aligns with the IMRaD requirements.

“I recommend putting the keywords in alphabetical order.”

The keywords were organized thematically, rather than alphabetically. We thought this arrangement provided better coherence for the reader. Nevertheless, we have made the adjustment and reordered them alphabetically.

“The Introduction lacks the broader context of the topic. This chapter contains only one reference! Although the formulations are acceptable, they are very general. I recommend that the authors place their work in a broader context, justify the need to investigate the topic and provide references.”

“The Introduction does not state what the specific objective is. Please, specify it accurately.”

“I recommend that the contributions be reworded and objectives given in the introduction.”

“I recommend that the structure of the manuscript be formulated at the end of the Introduction.”

We have revised the introduction to explicitly state the article's objective and outline the contributions made. Additionally, we have included the article's structure at the end of the introduction.

“Right at the beginning, I note that the manuscript contains only 23 references, which is a clear sign that the manuscript is not sufficiently processed from a scientific point of view and does not meet the basic requirements for scientific works.”

We believe that the quality of a scientific article cannot solely be measured by the quantity of references it contains. It's worth noting that some of the provided references are comprehensive surveys that analyze a substantial number of articles, hence the volume of reviewed material is significant. Additionally, we would like to emphasize the challenge in finding related works, given that the methodology for transitioning from simulation to real-world applications in aerial robotics is not commonly studied. Furthermore, the two proposed use cases are explored in a field where, as outlined in the related work, autonomous inspection is scarcely addressed within such facilities. Despite these challenges, we have diligently incorporated relevant references that offer valuable context to our research.

“The Methodology chapter is completely lacking in contextualization, it does not contain a single reference!”

In addition to the contextualization provided in the related work, we have included references to the methodology. However, as mentioned earlier, it is challenging to locate relevant articles in this particular field.

“A summary of the specific results is missing from the end of the manuscript. The wording of the Conclusions chapter is excessively general. Please, reconstruct, complement and specify it.”

We have included an experiments chapter. Within it, we present certain metrics that enable the validation of the proposed methodology's applicability. Furthermore, we assess the offsets between simulation and reality. However, as stated in the conclusions, future work will focus on minimizing this disparity. It's important to note that, given that this isn't a control article, the significance of the presented data lies in the resemblance between various developmental stages rather than the absolute values themselves.

“Overall, the manuscript contains important and relevant information and research, but in its current form it is too poor to be published as a scientific publication in this journal.”

We have paid special attention to the formal aspects of the paper, including clarity of presentation, organization, and adherence to formatting guidelines. We believe that these changes have significantly improved the manuscript's overall quality.

While we understand your concern about the importance of well-structured and professionally presented work, we also want to emphasize our shared goal of advancing scientific understanding in our field. We believe that the content of the manuscript offers valuable insights and relevant research that can contribute to the progress of our area of study.

In light of the improvements we've made, we kindly request your consideration for the publication of our manuscript. We are confident that the changes have addressed the concerns you raised while retaining the importance of the scientific content.

Once again, we thank you for your meticulous review and valuable input. We are grateful for the opportunity to enhance our work through your guidance.

Round 2

Reviewer 1 Report

Thank you for taking my suggestions into consideration and for the additional chapter. I think it completes the article.

Reviewer 2 Report

With the modifications have been made, the article became of acceptable quality for publication.

Quality of English is acceptable.